# Predictors of Psychological Strain and Allostatic Load in Teachers: Examining the Long-Term Effects of Biopsychosocial Risk and Protective Factors Using a LASSO Regression Approach

**DOI:** 10.3390/ijerph20105760

**Published:** 2023-05-09

**Authors:** Alexander Wettstein, Gabriel Jenni, Ida Schneider, Fabienne Kühne, Martin grosse Holtforth, Roberto La Marca

**Affiliations:** 1Department of Research and Development, University of Teacher Education Bern, 3012 Bern, Switzerland; 2Clinical Psychology and Psychotherapy, Department of Psychology, University of Bern, 3012 Bern, Switzerland; 3Psychosomatic Medicine, Department of Neurology, Inselspital, Bern University Hospital, 3010 Bern, Switzerland; 4Clinica Holistica Engiadina, Centre for Stress-Related Disorders, 7542 Susch, Switzerland; 5Clinical Psychology and Psychotherapy, Department of Psychology, University of Zurich, 8050 Zurich, Switzerland

**Keywords:** teacher stress, risk and protective factors, psychological strain, allostatic load, LASSO regression

## Abstract

Teacher stress significantly challenges teachers’ health, teaching quality, and students’ motivation and achievement. Thus, it is crucial to identify factors that effectively prevent it. Using a LASSO regression approach, we examined which factors predict teachers’ psychological strain and allostatic load over two years. The study included 42 teachers (28 female, *M*age = 39.66, *SD* = 11.99) and three measurement time points: At baseline, we assessed teachers’ (a) self-reports (i.e., on personality, coping styles, and psychological strain), (b) behavioral data (i.e., videotaped lessons), and (c) allostatic load (i.e., body mass index, blood pressure, and hair cortisol concentration). At 1- and 2-year follow-ups, psychological strain and allostatic load biomarkers were reassessed. Neuroticism and perceived student disruptions at baseline emerged as the most significant risk factors regarding teachers’ psychological strain two years later, while a positive core self-evaluation was the most important protective factor. Perceived support from other teachers and the school administration as well as adaptive coping styles were protective factors against allostatic load after two years. The findings suggest that teachers’ psychological strain and allostatic load do not primarily originate from objective classroom conditions but are attributable to teachers’ idiosyncratic perception of this environment through the lens of personality and coping strategies.

## 1. Introduction

Teachers report higher levels of self-perceived workplace stress [1,2] and higher rates of burnout [3] compared to other professions. According to the transactional stress model [4], an acute stress response emerges when a person appraises environmental demands as greater than their ability to meet, mitigate, or alter them [5,6]. Thus, the situation is appraised as potentially threatening, resulting in psychological, physiological, and behavioral stress responses [7,8].

Physiologically, acute stress prepares the body for an upcoming challenge and is not harmful if it lasts only for a limited time [9]. However, repeated or enduring stress exposure without adaptation can be harmful. Chronic stress can lead to long-term detrimental consequences for teachers physiologically (e.g., altered autonomic nervous system activity, hypothalamic–pituitary–adrenal axis dysfunction, or subclinical inflammation), psychologically (e.g., low frustration tolerance, depersonalization, or concentration problems), psychiatrically or psychosomatically (e.g., sleep disorders, depressiveness, or anxiety; [10,11]), and socially (e.g., withdrawal or social insecurity; [12]). Chronic stress may endanger health and well-being, jeopardize teaching quality [13], undermine students’ satisfaction and academic achievement [14], lead to high teacher turnover [15], and ultimately bring about high economic costs [16]. Thus, it is essential to identify risk and protective factors to prevent chronic stress in teachers.

However, linear stress models, such as the transactional stress model, neglect the complex and cumulative nature of stress exposure and cannot account for chronic stress [17]. The transdisciplinary stress model [17] goes beyond the acute stress response and conceptualizes stress as an emergent process in which a person’s contextual factors influence acute stress responses. Allostatic load results if acute stress responses are maladaptive (e.g., heightened anticipation or reactivity, or lack of habituation or recovery in regard to a stressor). Conversely, neural and peripheral changes that emerge from chronic stress can influence an individual’s life context. Contextual factors include individual-level (e.g., genetic or developmental) or environmental factors, cumulative stress exposure, and protective factors. Additionally, habitual processes, such as mental filters, can result from these contextual factors and, in turn, influence acute stress responses. Thus, this model (a) considers physiological processes, (b) extends the temporal dimension from acute to chronic stress, and (c) considers the influence of earlier stress experience on habitual cognition and acute stress responses.

Current research on teacher stress mainly focuses on teachers’ self-reporting, is often limited to one specific outcome, or is studied cross-sectionally. There has been less research that has examined interrelations among different stress-related outcomes, teachers’ stress physiology, objectifiable variables of real-life work experience, and longitudinal processes. Accordingly, the present longitudinal study explores the associations between different contextual factors in terms of predicting psychological strain and allostatic load over the course of two years.

Many possible factors influence whether teachers’ acute stress response is maladaptive, thus increasing their risk for chronic stress. In a literature review, Chang [18] distinguished three primary sources of teacher stress: (1) individual factors (e.g., personality and coping styles), (2) organizational factors (e.g., student misbehavior), and (3) transactional factors, which refer to interactions between individual and organizational factors (e.g., teachers’ judgments of student misbehavior and perceived social support from other teachers).

However, while many influencing factors are known, their common contribution to different chronic stress indicators has, to our knowledge, never been studied together. The present study aims to test essential contextual factors for risky or protective qualities regarding long-term stress consequences. We assume individual, organizational, and the resulting transactional factors to influence teachers’ acute stress response and, thus, affect long-term allostatic load indicators. Additionally, we also include measures of psychological strain, such as vital exhaustion, work overload, or problems in various life areas, as chronic stress indicators (Figure 1).

To determine relevant stress predictors, we use an innovative method of analysis, the least absolute shrinkage and selection operator (LASSO) approach to regression, which avoids the typical overfitting of standard regression models by performing regulation and selection. This procedure enables us to identify crucial risk and protective factors and examine their associations with psychological strain and allostatic load two years later while simultaneously considering many potential determinants. Importantly, the approach allows for a targeted selection of variables crucial for teacher stress research and indicates where prevention and intervention should start.

### 1.1. Individual Factors

Teachers’ stress response is influenced by their personality and their individual strengths and weaknesses [19]. As such, it is essential to identify potential individual risk and protective factors among these constructs.

*Core self-evaluation* (CSE) is seen as a protective factor against stress [20]. It describes a higher-order construct subsuming four well-known personality traits: global self-esteem, generalized self-efficacy, emotional stability, and internal locus of control [21]. People that score high in CSE see themselves as capable, worthy, and in control [22]. It has been proposed that CSE could be useful for understanding individual differences in stressor appraisal and response processes [23]. Some studies suggest that CSE may act as a protective factor by lowering an individual’s risk of burnout [24,25,26,27]. Furthermore, CSE is key to preventing stress consequences such as vital exhaustion (see below) among teachers [28].

More specifically, *teacher self-efficacy* refers to how capable teachers perceive themselves to be in the classroom, especially in challenging situations. It is negatively associated with teacher stress and burnout [29] and positively associated with students’ motivation [30].

In contrast, *neuroticism* (low emotional stability) is a central personality trait found to be an important risk factor for teacher stress [12,31] and refers to the disposition to experience negative affectivity, including anger, anxiety, self-consciousness, irritability, emotional instability, and depression [32]. People with elevated neuroticism react more strongly to environmental stress and interpret ordinary situations as threatening [33]. Kammeyer-Mueller et al. [23] suggest that emotional stability uniquely influences the stress processes, as compared to assessing CSE alone.

In addition to these essential personality constructs, *satisfaction* is a central protective factor. The concept is multifaceted, encompassing satisfaction with work, family, friends, oneself, and life in general. Individuals with high satisfaction levels tend to view things more optimistically and actively create a more favorable environment, supported by more positive attitudes [34]. While CSE is associated with higher job and life satisfaction [35,36], individuals with high levels of neuroticism often exhibit lower levels of life satisfaction [37].

Furthermore, teachers can use different strategies to cope with challenging situations. The suitability of a coping strategy usually depends on situational demands, and using strategies flexibly is considered adaptive. However, individuals can habitually adopt certain coping styles, which can be categorized as approach or avoidance strategies. *Approach strategies* are generally seen as adaptive coping styles and include active coping strategies such as active problem-solving or seeking social support. These strategies have been found to be adaptive, especially in the medium and long term, and are negatively related to emotional exhaustion [38]. However, overcommitment might endanger teachers’ health because the teaching profession is characterized by fuzzy boundaries (i.e., one can always do more and try to improve). Accordingly, an excessive readiness to expend time and energy as well as striving for perfection are risk factors for experiencing high levels of stress, while the ability to set limits and temporarily disengage might function as a protective factor in the teaching profession. *Avoidance strategies* such as resignation or excessive social withdrawal are viewed as risk factors because the avoided problems remain unresolved. In turn, an unresolved problem can continue to generate stress, thus reducing stress resilience in the long term. Hence, avoidant coping strategies are likely to deplete resources and contribute to stress while increasing emotional exhaustion over time [39].

People with high CSE levels might choose more adaptive coping styles and use them more effectively. A meta-analysis by Kammeyer-Mueller et al. [23] showed a positive relationship between CSE and problem-solving coping, whereas CSE and avoidant coping were negatively associated. Additionally, maladaptive coping strategies used by teachers are negatively related to life satisfaction [40].

### 1.2. Organizational Factors

Organizational factors refer to objective working conditions, including workload (e.g., total number of lessons taught per week) and objective workplace characteristics (e.g., the classroom environment). Teachers are confronted with a high density of interactions and have to simultaneously manage processes of teaching and learning as well as social interactional processes. Therefore, it is essential to not only focus on the teacher but to also consider the risk and protective factors in the teacher’s environment.

Classroom disruptions are considered a critical source of teacher stress [41,42]. They include non-aggressive student disruptions, henceforth described as *student disruptions* (e.g., agitation or cutting in), and aggressive student behavior, henceforth described as *student aggression* (e.g., threatening, shaming, or ridiculing), which is defined as any behavior intended to harm another person or destroy property [43]. Classroom disruptions can affect the entire methodological-didactic setting, termed *setting disruptions*, and can lead to a working atmosphere marked by interruptions, a lack of concentration, and restlessness [44].

Effective *classroom management* prevents classroom disruptions. Classroom management includes all “actions teachers take to create an environment that supports and facilitates both academic and socio-emotional learning” [45] (p. 4). Additionally, good *teacher–student relationships* are a prerequisite for a good classroom environment. Teachers’ efficacy in classroom management and good teacher–student relationships are both regarded as key protective factors not only for teachers’ health [46,47], but also for students’ psychosocial development, motivation, and learning success [48,49].

### 1.3. Transactional Factors

Besides objective factors of the classroom environment, teachers’ perception of this environment against the background of their unique personalities and coping styles plays a decisive role. Teachers’ perception of the classroom environment is transactional in nature, resulting from the interplay between individual and organizational factors. Appraisals of perceived environmental demands and available resources are key elements of the transactional stress theory by Lazarus and Folkman [4], with appraised resources counteracting appraised demands.

Studies have revealed only moderate associations between observed and teacher-perceived classroom environments [50,51,52]. Teachers perceive student aggression through their own subjective filter. For instance, personality traits such as neuroticism and dysfunctional coping styles are associated with increased perceptions of aggressive student behavior [53,54,55]. Specifically, teachers with high resignation tendencies and chronic worry seem to systematically overestimate aggressive student behavior in their classrooms [56]. Thus, teacher stress is not exclusively an objective reflection of external stressors, but also mirrors teachers’ personalities and coping styles [57]. As such, it is important to assess the classroom environment through both objective observation and from the teacher’s own perspective.

In addition, social factors can influence teachers’ stress experience. Social interactions are fundamental to the teaching profession. To successfully master challenging social interactions in everyday school life, teachers depend on functional cooperation within multiprofessional teams. As with the classroom environment, the teacher’s perception plays an essential role. Perceived social support from other teachers, school management, or administrators is vital and can protect against stress. Therefore, teachers need to feel sufficiently supported by their colleagues and the school management/administration. Social support includes instrumental and emotional support and can protect teachers from burnout [58].

Conversely, perceived negative social interactions can also contribute to teacher stress. Risk factors such as fear of negative evaluation, a lack of social recognition, social isolation, or social tensions contribute to teacher anxiety and burnout [18]. *Social tensions* (e.g., disagreements, conflicts, and differing role expectations) in a team constitute a risk factor for teacher stress. Moreover, *social isolation* (i.e., the absence of social relationships and interactions with family members, friends, or peers; [59]) is linked to neurobiological processes that activate the organism’s stress response, leading to hypervigilance to social threats such as rejection, exclusion, negative evaluation, and feeling unsafe [60,61]. Furthermore, a *lack of social recognition* and the personality factor *fear of negative evaluation*, which is highly relevant for social interactions, might exacerbate teacher stress.

### 1.4. Teachers’ Psychological Strain and Allostatic Load

Long-term stress exposure can result in a variety of psychological and physiological costs that have a profound impact on the affected individual. While chronic stress is often caused by work, it also affects life in general, such as dealing with the demands of family life or personal challenges. Prolonged stress exposure hampers physiological regulation and may lead to allostatic overload [62]. Such physiological consequences of stress might go unnoticed. To fully understand teacher stress, it is therefore essential to assess not only work-related symptoms of stress and burnout but also a variety of psychological and physiological consequences.

#### 1.4.1. Psychological Strain

Individuals dealing with long-term stress can experience various adverse psychological consequences, such as vital exhaustion. *Vital exhaustion* refers to a psychosomatic state of unusual fatigue, a lack of energy, irritability, and demoralization. It is considered a potential early warning sign of cardiovascular disease [63] and is closely related to burnout [64].

Ongoing stress can also lead to *chronic worry*, which results from insufficient adaptive emotion regulation strategies meant to temporarily avoid physiological arousal and negative emotions [65]. Accordingly, individuals who frequently worry show higher levels of physiological arousal before and after a stress-inducing situation [66].

A further consequence of ongoing occupational stress is *work overload* [67]. In a sample of 166 teachers, “workload” was among the most cited responses to the question of what is most stressful in life, second only to “work” [40]. Two meta-analyses on stress in teachers revealed a strong association between work overload and burnout [68,69].

Psychological problems and strain can manifest in different areas of life. *Occupational problems* include negative feelings related to work; *physical problems* refer to medical issues such as headaches, sleep problems, recurrent infections, or gastrointestinal symptoms; *self-related problems* include a low frustration tolerance, feelings of depersonalization, or concentration problems; and *family-related problems* encompass estrangement or decreased participation in family life.

#### 1.4.2. Allostatic Load

Besides the psychological consequences, chronic stress also manifests physiologically. Multisystemic physiological dysregulation might result in allostatic overload [62] and affect teachers’ ability to adapt and effectively respond to the environment. Specifically, body mass index, blood pressure, and cortisol are biomarkers that reflect a higher allostatic load [62]. Stress is positively associated with body mass index (BMI; kg/m^2^) and adiposity [70,71]. For instance, Harding et al. [72] found that psychosocial stress (perceived stress and stressful life events) led to weight gain over a period of five years.

Other standard markers of physiological stress are *systolic blood pressure* (*SBP*) and *diastolic blood pressure* (*DBP*). Chronic stress is often associated with hypertension [73]. In particular, perseverative thinking, such as chronic worry, has been linked to increased blood pressure [74].

Furthermore, chronic stress can also be assessed through hair cortisol. *Hair cortisol concentration* (*HCC*) is a relatively new biomarker of long-term cumulative chronic physiological stress exposure [28,75], with the amount of cortisol in the hair providing information about how much a person is burdened by persistent stress. The longer stress lasts, the longer an increased concentration of cortisol circulates in the body, resulting in an increased accumulation of cortisol in the hair.

### 1.5. The Present Study

Overall, teachers’ stress experience can have manifold influences, leading to psychological or physiological stress consequences. However, studies often include only a few factors that influence chronic stress. Less is known about the interplay between these variables. Moreover, most studies solely rely on self-reports. However, combining questionnaires and observational data to validate teachers’ perception is important. Additionally, studies that assess both long-term psychological and physiological stress show inconsistent results on whether the two are associated.

The present study aims to fill these gaps by examining the interplay between different predictors and long-term outcomes of stress. We combine self-reports and behavioral observation and include psychological and physiological measures of stress consequences. This allows the selection of essential protective factors and, ultimately, the development of effective teacher stress prevention programs. We explored which risk and protective factors predict teachers’ psychological strain and allostatic load two years after a baseline assessment, examining the following research questions:How stable are psychological strain and allostatic load over a period of two years (1a), and how high are the intercorrelations between the different consequences of stress (1b)?Which factors (individual, organizational, or transactional) at baseline predict the different indicators for psychological strain (2a) and allostatic load (2b) two years later?

To identify predictors of stress, we employed the LASSO method, using leave-one-out cross-validation (LOOCV). LASSO is a method of regression analysis that performs variable selection and regularization [76], which improves the predictive accuracy and interpretability of the resulting statistical model. The utility of LASSO regression for predictor selection has only recently gained attention and is an emerging trend [77]. We chose to use LASSO regression and LOOCV in the present study as this method is suitable for estimating models with multiple predictors in a small sample [78], avoiding overfitting the data and lowering test error bias [79].

As such, this study contributes to the existing literature by studying various possible factors influencing long-term psychophysiological stress consequences, using LASSO to determine the best predictors for each chronic stress indicator. This will allow future studies to select important variables, thereby improving the accuracy and economy of models. In addition, it adds to our understanding of long-term teacher stress and allows for targeted prevention and intervention.

## 2. Materials and Methods

This study is part of a larger project on the biopsychology of teacher stress [28,80].

### 2.1. Participants

The study included 42 teachers (28 female, *M*age = 39.66, *SD* = 11.99) at baseline. After the first measurement, 1 teacher moved abroad and 2 teachers withdrew their participation due to pregnancy, resulting in a sample of 39 teachers two years later. Participants were recruited via flyers and circular emails. Inclusion criteria for participation in the study were employment as a primary or secondary teacher in the Swiss canton of Bern and a workload of at least 16 lessons per week (equivalent to at least 60 percent of full-time employment). Exclusion criteria were working outside of the canton of Bern, acute infections, cardiovascular or other chronic diseases, use of cardiovascular drugs or other medication in the past two months (except phytopharmaceuticals), substance abuse, consumption of psychoactive substances in the last four weeks, more than two standard units of alcohol per day, smoking more than ten cigarettes per day, long-distance flights within the last two weeks, and pregnancy. All teachers were screened for the inclusion and exclusion criteria during a short telephone interview. Enrolled participants provided informed consent. The study was approved by the ethics committee of the canton of Bern and by the Internal Review Board (IRB) of the University of Bern, and was conducted in strict compliance with current data protection laws and in accordance with the Declaration of Helsinki.

### 2.2. Measures and Design

Assessments were conducted at three time points, i.e., baseline (t0), 1-year follow-up (t1), and 2-year follow-up (t2). At the first measurement time point (t0), we assessed teachers’ (a) self-reports (i.e., questionnaires on teachers’ personality, coping styles, and psychological strain), (b) behavioral data (i.e., 202 videotaped lessons coded by trained external observers), and (c) allostatic load (i.e., BMI, DBP, SBP, and HCC). At the 1- and 2-year follow-ups, we reassessed psychological strain and allostatic load biomarkers. Only physical problems and HCC were not assessed at the final measurement time point (t2).

#### 2.2.1. Self-Reports

Independent and dependent variables were measured using established instruments. An overview of all measured variables, the instruments used, and their internal consistency is presented in Table 1 (predictors) and Table 2 (outcomes).

#### 2.2.2. Behavioral Observation

For the behavioral observation of the classroom environment, GoPro cameras and microphones were installed in the classroom of each teacher. Four trained observers, who had previously reached a criterion of an interrater agreement of 0.80 (Cohen’s kappa), coded student aggression and setting disruptions in an event sampling procedure using the Observation System BASYS [87]. Setting disruptions included a working atmosphere marked by interruptions, a lack of concentration, and restlessness. Student aggression comprised any behavior intended to harm another person or to destroy property, with a distinction drawn between verbal and physical aggressive behavior and between direct (e.g., insulting or hitting) and indirect (e.g., hiding objects or spreading false rumors) aggressive behavior. Higher values on the BASYS represent higher numbers of disruptions and greater aggressive behavior. In addition, teacher–student relationships and classroom management were rated using the Classroom Questionnaire [51], with higher scores representing better relationships and better management of the challenges faced in the classroom.

#### 2.2.3. Physiological Measures

*BMI* was calculated by dividing the teachers’ weight (kg) (Seca 813; Reinach, Switzerland) by the square of their height (m) (Seca 213; Reinach, Switzerland).

Resting blood pressure was measured using an OMRON HBP-1120 (Kyoto, Japan), which is a fully portable, non-invasive blood pressure device that measures systolic and diastolic blood pressure (SBP and DBP). The blood pressure cuff was applied to each participant’s non-dominant arm.

*Hair cortisol concentration (HCC)* was determined from a hair sample collected from the posterior vertex region of the head, using the 3 cm segment closest to the scalp. Given an average hair growth of 1 cm per month, this segment represents the cumulative glucocorticoid secretion over the three months before sampling [92]. HCC was measured using liquid chromatography–tandem mass spectrometry (LC–MS/MS) [93].

### 2.3. Data Analyses

The Shapiro–Wilk test was used for each variable to test whether the data were normally distributed. Variables not fulfilling the criteria of normal distribution were log-transformed. All correlations and regressions were calculated and interpreted with these transformed variables included. Descriptive statistics and bivariate Pearson correlations were computed to investigate the test–retest reliability and intercorrelations of the stress-related consequences at follow-up.

In the second step, we applied the LASSO method to select predictors from the list (Table 1; 40 predictors) using LOOCV (38 folds). LASSO regressions helped to identify the best combination of predictors that explained teachers’ psychological strain and allostatic load at 1- and 2-year follow-up (Table 2). All variables were standardized to be on the same scale of influence on the penalty term (λ) in the LASSO regression. For several dependent variables, the number of regularized coefficients >0 calculated by the LASSO regression was more than 10. Based on the variance importance output from the LASSO regression, we chose only predictors that accounted for at least 25% of the beta coefficients of the most important predictors. Variables with a share of less than 25% were not considered further. Since the whole sample (N = 39) was used for LOOCV, there was no validation set to obtain unbiased performance measures (MSE and R^2^). Consequently, no performance measures are reported in the results section. LASSO regressions were conducted using the caret [94] package in R and all descriptive statistics and bivariate correlations were calculated using SPSS version 28.

## 3. Results

### 3.1. Longitudinal Stability of Psychological Strain and Allostatic Load

Means, standard deviations, and bivariate correlations between the psychological strain variables at t0, t1, and t2 are presented in Table 3. Vital exhaustion (t2: *r* = 0.77), work overload (t2: *r* = 0.72), and physical problems (t1: *r* = 0.84) showed the highest longitudinal stability, while occupational problems (t2: *r* = 0.46), chronic worry (t2: *r* = 0.58), self-related problems (t2: *r* = 0.59), and family-related problems (t2: *r* = 0.41) showed more variation. At t1 and t2, self-related problems correlated highly with most other stress-related consequences.

Means, standard deviations, and bivariate correlations between the allostatic load biomarkers at t0, t1, and t2 are shown in Table 4. BMI (t2: *r* = 0.93) showed the highest longitudinal stability, followed by SBP (t2: *r* = 0.73), DBD (t2: *r* = 0.65), and HCC (t1: *r* = 0.52). BMI showed a significant positive correlation with DBP and a weaker, less significant correlation with SBP. No allostatic load variable was significantly related to HCC.

Bivariate correlations between allostatic load and psychological strain are presented in Table 5. BMI showed a trend of a negative correlation with many of the psychological stress outcomes apart from family-related problems, which was positively correlated with BMI. Most associations were not statistically significant. There was a significant negative correlation between SBP at t0 and vital exhaustion at t2 (*r* = −0.34) and a significant positive correlation between work overload at t0 and SBP at t2 (*r* = 0.36). Additionally, DBP at t0 was positively correlated with family-related problems at t1 (*r* = 0.37) and t2 (*r* = 0.42), and DBP at t2 was positively correlated with family-related problems at t2 (*r* = 0.40).

### 3.2. Predictors of Psychological Strain and Allostatic Load

Below, we present the regularized LASSO regression coefficients of the most important predictors at the first measurement (t0) for psychological strain and allostatic load at the 2-year follow-up (t2) and for physical problems and HCC at the 1-year follow-up (t1). An overview of all significant LASSO-regularized regression coefficients is presented in Table 6.

#### 3.2.1. Psychological Strain at Follow-Up

**Vital Exhaustion.** CSE (β = −0.23) emerged as the most important protective factor, showing a negative association with vital exhaustion. This was followed by the two risk factors of perceived setting disruptions (β = 0.11) and neuroticism (β = 0.08), which were both positively associated with vital exhaustion.

**Chronic Worry.** Chronic worry was most strongly associated with the risk factor of perceived student disruptions (β = 0.26), followed by the risk factor of neuroticism (β = 0.19) and the protective factor of CSE (β = −0.16), which predicted less worrying at follow-up.

**Work Overload**. The strongest predictor was the risk factor of neuroticism, which was positively associated with work overload (β = 0.20). In contrast, protective factors such as the coping style of seeking positive experiences (β = −0.19) and life satisfaction (β = −0.10) negatively predicted teachers’ work overload at follow-up. Additionally, the two risk factors of perceived student disruptions (β = 0.14) and excessive work engagement (β = 0.11) were positively associated with work overload at follow-up.

**Occupational Problems.** The strongest predictor was the protective factor of life satisfaction, which was negatively associated with occupational problems (β = −0.30) at follow-up. The most predictive risk factors were perceived student disruptions (β = 0.23), neuroticism (β = 0.18), and perceived setting disruptions (β = 0.10). It further emerged that CSE (β = −0.09) was also a significant protective factor relating to occupational problems at follow-up.

**Physical Problems.** The most strongly related protective factor regarding physical problems was the experience of social support (β = −0.20). In contrast, perceived student disruptions (β = 0.19) and (surprisingly) the assumed protective factor of perceived classroom management (β = 0.21) were positively associated with physical problems. With respect to coping strategies, physical problems were negatively related to the ability to distance oneself (β = −0.13) and positively related to resignation tendency (β = 0.09). Additionally, physical problems were negatively related to CSE (β = −0.13), satisfaction with oneself (β = −0.06), and offensive problem solving (β = −0.06) and positively related to neuroticism (β = 0.07).

**Self-Related Problems.** Perceived student disruptions were the strongest predictor of self-related problems (β = 0.15), with more disruptions leading to greater problems. This was followed by life satisfaction (β = −0.15), which reduced such problems. Neuroticism (β = 0.08) and CSE (β = −0.06) were also associated with self-related problems, with the former emerging as a risk factor and the latter a protective factor. Furthermore, self-related problems were inversely associated with the ability to distance oneself (β = −0.10).

**Family-Related Problems.** For family-related problems, the LASSO regression set all regularized coefficients to 0.

#### 3.2.2. Allostatic Load at Follow-Up

**BMI.** The strongest predictor of BMI was resignation tendency, (β = −0.19), followed by support from other teachers (β = −0.18), the risk factor of perceived student disruptions (β = 0.18), and the protective factor of perceived classroom management (β = −0.13). With regard to coping strategies, inner peace and balance (β = 0.17) and subjective meaningfulness of work (β = 0.07) were associated with BMI. The analyses further revealed associations with support from the school administration (β = −0.12), social tensions (β = 0.07), extraversion (β = 0.07), total number of lessons taught per week (β = −0.05), and observed student aggression (β = −0.05), again uncovering some unexpected associations.

**Systolic Blood Pressure.** Support from the school administration showed the strongest association with SBP (β = −0.17), predicting a lower SBP at follow-up. SBP was also associated, albeit less strongly, with support from other teachers (β = −0.10). This was followed by the ability to distance oneself (β = −0.11) and observed teacher–student relationships (β = −0.06), with higher values predicting a lower SBP.

**Diastolic Blood Pressure.** The most important predictor of DBP was support from the school administration (β = −0.14), with higher values predicting a lower DBP. This was followed by seeking positive experiences (β = −0.13) and support from other teachers (β = −0.11), which also predicted a lower DBP.

**Hair Cortisol**. For HCC, the LASSO regression set all regularized coefficients to 0.

## 4. Discussion

The present study examined how teachers’ personal characteristics, organizational factors, and transactional factors are associated with their psychological strain and allostatic load after two years. The study included a multimethod approach using teachers’ self-reports, behavioral observations, and physiological measures in a longitudinal design. Investigating a broad range of predictors, we explored their role in predicting both psychological and physiological stress-related outcomes using LASSO regressions. As such, we were able to investigate stability (1a) and intercorrelations (1b) of a variety of consequences of stress and examine the contribution of different predictors of teachers’ psychological strain (2a) and allostatic load (2b) from baseline to two years later.

Regarding question 1a, psychological strain showed high longitudinal stability over two years. In particular, vital exhaustion and work overload were highly stable, suggesting long-term problems regarding work–life balance. For teachers with pronounced vital exhaustion, this high longitudinal stability is especially alarming because vital exhaustion is a potential early warning sign of cardiovascular disease [63]. Additionally, there is evidence that vital exhaustion significantly predicts the reoccurrence of vascular events [95]. As expected, allostatic load indicators showed moderate to high longitudinal stability. Whereas BMI and SBP were highly stable, DBP and HCC showed only moderate stability. However, this could be due to our measurements, as we only measured blood pressure during a resting interval of five minutes [96].

Regarding question 1b, psychological strain variables showed considerable intercorrelations, suggesting that different facets of stress from different areas are strongly interrelated. In contrast, different allostatic load indicators showed only weak to moderate intercorrelations, suggesting that the different indicators reflect different physiological and behavioral processes. This is in line with findings from Juster and colleagues [62], who demonstrated that the allostatic load index can be broken down into two non-overlapping clusters: neuroendocrine biomarkers (e.g., HCC) versus metabolic syndrome biomarkers (e.g., blood pressure and BMI). Each of these clusters might contribute independently to health risks. Overall, psychological and physiological stress-related consequences were not associated, albeit with a few exceptions. This finding corresponds with a review of 49 studies examining the association between psychological and physiological stress responses, which only revealed significant correlations between acute cortisol responses and perceived emotional stress variables in approximately 25% of the studies [97].

Regarding question 2a, our results showed that psychological strain is primarily predicted by individual and transactional factors and no organizational factors. This is consistent with Lazarus and Folkman’s [4] transactional stress model, which posits that what matters is not so much what potential environmental stressors an individual is exposed to, but how individuals evaluate them in light of their own resources. Perceived student disruptions predicted a wide range of teachers’ psychological stress-related outcomes two years later. Noteworthily, CSE was a vital protective factor, affecting five out of seven psychological stress outcomes. In contrast, neuroticism proved to be a central risk factor, affecting six out of seven psychological outcomes two years later. This corresponds to previous studies showing that neuroticism is the strongest correlate of burnout [24,31,98]. Thus, teachers with high neuroticism might benefit from targeted interventions, and increasing their CSE could help tackle adverse effects.

Regarding question 2b, allostatic load was partly predicted by relatively unmediated organizational factors. In addition, whereas social factors such as support from other teachers or the school administration did not have a protective effect on psychological strain, they had a protective effect on teachers’ allostatic load. In line with this finding, previous research found that social support buffers the impact of stress on blood pressure [99], and support from other teachers and the school administration can reduce work pressure and increase experienced self-efficacy and general well-being [100]. Noteworthily, HCC was not predicted by any of our investigated variables. It can be assumed that self-reports are poor indicators of HCC because they are based on higher cognitive processing.

Overall, while not all factors seem essential to predicting long-term teacher stress, some key factors, such as neuroticism, CSE, or perceived student disruptions, predicted a variety of stress outcomes. Interestingly, psychological strain was predicted by solely individual and transactional (i.e., more subjective) factors, whereas predictors for allostatic load were more of an organizational nature. Further, psychological strain and allostatic load were not or were only weakly correlated. Individuals construct their view of the world through numerous attributional processes. From a systems theory perspective, a person consists of two systems, an evolutionary older biological system and a younger psychological system [101]. Both systems influence each other only to a small degree, or they co-evolve [102]. Physiological processes based on the evolutionarily older system respond more immediately to environmental stressors and are much less amenable to being influenced by the conscious cognitive system. Accordingly, it is challenging for the psychological system to interpret the biological processes of its own body correctly [102,103]. Unfavorable physiological stress responses, such as high blood pressure or chronically increased cortisol exposure, are often unrecognized. Thus, it is essential to capture not only teachers’ subjective experience but also more objectifiable factors of their classroom environment. Teachers should be sensitized to physiological processes as they might not notice harmful consequences because they do not correlate with psychological stress outcomes. In addition, this could also lead to misattribution of physiological symptoms, such that organizational factors (e.g., number of lessons) as a source are not detected, and thus, not addressed. However, our results also show that support within the working environment (i.e., from other teachers and the administration) protects against physiological long-term stress consequences.

The present findings are limited by the small sample size and the fact that our sample consisted of apparently healthy and medication-free teachers. Thus, the results should not be generalized to the entire population of teachers. Furthermore, as the teachers taught different classes over the duration of the study, longitudinal conclusions regarding the classroom environment should be treated with caution. Moreover, we measured blood pressure only once during a resting interval. A 24 h measurement would have yielded a more comprehensive picture.

Nevertheless, this study also has some considerable strengths. The combination of self-reports with observational and physiological data improves our understanding of the interplay between psychological, behavioral, and physiological processes. A further strength lies in the longitudinal design with a relatively large number of predictors. Additionally, the longitudinal data were analyzed using the innovative LASSO regression approach, which is suitable for small sample sizes and a large number of predictors.

Future research should examine whether these longitudinal findings can be replicated in larger or clinical samples. Moreover, each psychological or physiological stress-related consequence, with its associated predictors found in this study, represents an individual model that needs to be replicated. Accordingly, the present paper provides an initial orientation about which variables are likely associated with each consequence of stress, especially over time.

Future studies should also include recursive processes (i.e., the influence of chronic stress on the acute stress experience). These processes, although included in our model, were not the focus of the present study. However, there is evidence that high chronic psychological stress affects teacher perceptions. For example, a study from the same project showed that chronically worried teachers systematically overestimated the extent of student aggression in their class [56].

In summary, our preliminary study of 40 predictors and 11 outcomes provides an overall picture of how different predictors may be uniquely associated with different psychological and physiological stress outcomes. The chosen approach not only facilitates the planning of future research in terms of variable selection, but also provides guidance on key factors for the development of prevention programs.

## 5. Conclusions

The present study demonstrates that vital exhaustion and work overload remain very stable over two years and that teachers’ psychological strain and allostatic load are barely associated. Teachers’ psychological strain is primarily associated with the two main risk factors, neuroticism and perceived student disruptions, and might be prevented by developing a positive core self-evaluation. By contrast, a high allostatic load might best be prevented by social support from other teachers and school administrators. In general, the findings highlight the importance of supporting teachers by strengthening their resources and, if necessary, raising their awareness of individual risk factors and unfavorable coping strategies to help them remain healthy in their challenging profession and continue to teach well.

## Figures and Tables

**Figure 1 ijerph-20-05760-f001:**
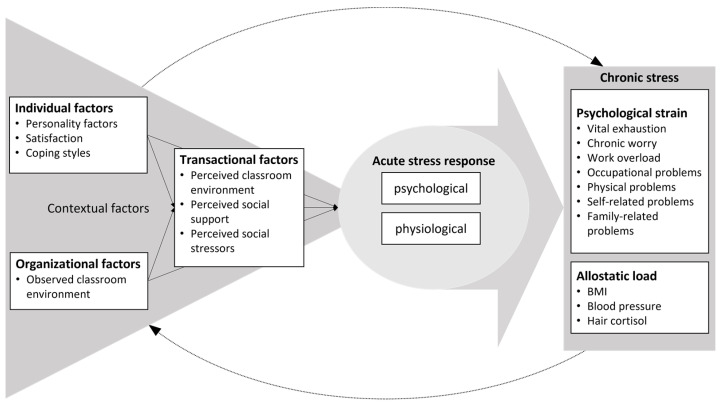
Transdisciplinary stress model including individual, organizational, and transactional factors that influence teachers’ acute stress response. When acute stress responses are ineffective, chronic stress, measured psychologically (i.e., psychological strain) or physiologically (i.e., allostatic load), can result.

**Table 1 ijerph-20-05760-t001:** Overview and consistency of the predictor variables.

	Variable	Test	Items	Alpha
Individual factors	Core self-evaluation	CSE [81]	12	0.81
	Teacher self-efficacy	[82]	4	0.67
	Neuroticism	BFI-10 [83]	4	0.74
	Extraversion	BFI-10	2	0.84
	Personal competence	RS-11 [84]	9	0.77
	Satisfaction			
	Life satisfaction	AVEM [85]	6	0.81
	Work satisfaction	BOSS [86]	5	0.65
	Satisfaction with oneself	BOSS	5	0.70
	Satisfaction with family	BOSS	5	0.98
	Satisfaction with friends	BOSS	5	0.89
	Coping styles			
	Excessive work engagement	AVEM	6	0.84
	Striving for perfection	AVEM	6	0.88
	Sense of achievement at work	AVEM	6	0.76
	Professional ambition	AVEM	6	0.87
	Significance of work	AVEM	6	0.81
	Resignation tendency	AVEM	6	0.84
	Offensive problem solving	AVEM	6	0.80
	Ability to distance oneself	AVEM	6	0.86
	Inner peace and balance	AVEM	6	0.69
	Seeking positive experiences	Own scale; unpubl.	2	0.73
Organizational factors	Total lessons		1	
	Observed classroom environment			
	Student disruptions	CQ [51]	-	-
	Student aggressions	BASYS [87]	-	-
	Setting disruptions	CQ		
	Classroom management	CQ	-	-
	Teacher–student relationship	CQ	-	-
Transactional factors	Perceived classroom environment			
	Student disruptions	CQ	4	0.77
	Student aggressions	CQ	4	0.91
	Setting disruptions	CQ	4	0.71
	Classroom management	CQ	3	0.76
	Teacher–student relationship	CQ	6	0.73
	Perceived social support			
	Experience of social support	AVEM	6	0.83
	Instrumental support	BSSS [88]	4	0.80
	Support from other teachers	Own item; unpubl.	1	-
	Support from school administration	Own item; unpubl.	1	-
	Perceived social stressors			
	Fear of negative evaluation	SANB5 [89]	5	0.88
	Lack of social recognition	TICS [90]	4	0.77
	Social isolation	TICS	6	0.83
	Social tensions	TICS	6	0.91

Note. 38 independent variables. CSE = Core Self-Evaluations Scale; BFI-10 = Big Five Inventory 10-items; RS-11 = Resilience Scale; AVEM = Measure of Coping Capacity Questionnaire; BOSS = Burnout Screening Scales; BASYS = Observation System for the Analysis of Aggressive Behavior; CQ = Classroom Questionnaire; BSSS = Berlin Social Support Scales; SANB5 = Fear of Negative Evaluation Scale; TICS = Trier Inventory for Chronic Stress.

**Table 2 ijerph-20-05760-t002:** Overview and consistency of the dependent psychological strain variables.

Variable	Test	Items	Cronbach’s Alpha		
			t0	t2	t3
Vital exhaustion	MQ [91]	21	0.88	0.88	0.91
Chronic worry	TICS [90]	4	0.92	0.90	0.90
Work overload	TICS	8	0.95	0.94	0.95
Occupational problems	BOSS [86]	10	0.91	0.83	0.83
Physical problems	BOSS	10	0.74	0.71	-
Self-related problems	BOSS	10	0.91	0.87	0.91
Family-related problems	BOSS	5	0.81	0.86	0.84

Note. MQ = Maastricht Vital Exhaustion Questionnaire; TICS = Trier Inventory for Chronic Stress; BOSS = Burnout Screening Scales.

**Table 3 ijerph-20-05760-t003:** Descriptive statistics and intercorrelations of psychological strain at t0, t1, and t2.

Variable	1	2	3	4	5	6	7	8	9	10	11	12	13	14	15	16	17	18	19	20
1. VE t0	--																			
2. VE t1	0.78 **	--																		
3. VE t2	0.77 **	0.74 **	--																	
4. CW t0	0.74 **	0.58 **	0.60 **	--																
5. CW t1	0.67 **	0.68 **	0.57 **	0.75 **	--															
6. CW t2	0.62 **	0.62 **	0.77 **	0.58 **	0.75 **	--														
7. WO t0	0.80 **	0.62 **	0.49 **	0.72 **	0.65 **	0.49 **	--													
8. WO t1	0.69 **	0.75 **	0.55 **	0.61 **	0.67 **	0.59 **	0.74 **	--												
9. WO t2	0.65 **	0.64 **	0.68 **	0.48 **	0.52 **	0.66 **	0.72 **	0.72 **	--											
10. OP t0	0.78 **	0.60 **	0.47 **	0.63 **	0.53 **	0.35 **	0.75 **	0.61 **	0.44 **	--										
11. OP t1	0.64 **	0.59 **	0.53 **	0.54 **	0.68 **	0.59 **	0.61 **	0.76 **	0.57 **	0.59 **	--									
12. OP t2	0.63 **	0.60 **	0.74 **	0.56 **	0.64 **	0.74 **	0.58 **	0.68 **	0.79 **	0.46 **	0.76 **	--								
13. PP t0	0.69 **	0.62 **	0.67 **	0.65 **	0.65 **	0.70 **	0.57 **	0.53 **	0.54 **	0.51 **	0.50 **	0.61 **	--							
14. PP t1	0.70 **	0.73 **	0.68 **	0.60 **	0.64 **	0.65 **	0.56 **	0.57 **	0.53 **	0.55 **	0.59 **	0.64 **	0.84 **	--						
15. SP t0	0.88 **	0.65 **	0.61 **	0.68 **	0.68 **	0.53 **	0.78 **	0.62 **	0.59 **	0.83 **	0.59 **	0.58 **	0.72 **	0.66 **	--					
16. SP t1	0.61 **	0.76 **	0.53 **	0.48 **	0.71 **	0.62 **	0.57 **	0.77 **	0.65 **	0.55 **	0.77 **	0.70 **	0.60 **	0.72 **	0.65 **	--				
17. SP t2	0.59 **	0.58 **	0.80 **	0.47 **	0.58 **	0.79 **	0.48 **	0.56 **	0.78 **	0.39 *	0.59 **	0.85 **	0.63 **	0.65 **	0.59 **	0.68 **	--			
18. FP t0	0.56 **	0.45 **	0.45 **	0.47 **	0.45 **	0.54 **	0.59 **	0.56 **	0.55 **	0.61 **	0.51 **	0.59 **	0.47 **	0.52 **	0.64 **	0.65 **	0.60 **	--		
19. FP t1	0.45 **	0.46 **	0.25	0.34 *	0.42 **	0.33 *	0.54 **	0.55 **	0.51 **	0.48 **	0.38 *	0.39 *	0.50 **	0.51 **	0.57 **	0.70 **	0.41 *	0.73 **	--	
20. FP t2	0.07	−0.03	0.17	0.14	0.12	0.34 *	0.12	0.22	0.33 *	−0.07	0.17	0.24	0.13	0.04	0.11	0.21	0.35 *	0.41 *	0.40 *	--
*M*	31.48	32.23	31.36	8.60	8.26	8.13	21.83	19.15	20.67	17.14	16.72	17.59	16.55	16.08	17.98	17.59	18.82	10.55	9.74	10.46
*SD*	8.71	7.95	9.44	3.80	3.57	3.06	7.25	7.04	6.84	7.63	4.84	5.79	5.63	5.04	8.06	5.71	7.90	4.51	3.91	4.31

Note. t0: N = 42; t1: N = 39. * *p* < 0.05, ** *p* < 0.01, two-tailed. VE = vital exhaustion; CW = chronic worry; WO = work overload; OP = occupational problems; PP = physical problems; SP = self-related problems; FP = family-related problems.

**Table 4 ijerph-20-05760-t004:** Descriptive statistics and intercorrelations of physiological variables at t0, t1, and t2.

Variable	1	2	3	4	5	6	7	8	9	10	11
1. BMI t0	--										
2. BMI t1	0.96 **	--									
3. BMI t2	0.93 **	0.97 *	--								
4. DBP t0	0.39 **	0.38 *	0.38 *	--							
5. DBP t1	0.33 *	0.33 *	0.33 *	0.65 **	--						
6. DBP t2	0.40 *	0.38 *	0.40 *	0.65 **	0.81 **	--					
7. SBP t0	0.30	0.27	0.26	0.82 **	0.60 **	0.51 **	--				
8. SBP t1	0.34 *	0.34 *	0.36 *	0.63 **	0.85 **	0.67 **	0.75 **	--			
9. SBP t2	0.27	0.25	0.29	0.69 **	0.78 **	0.87 **	0.73 **	0.79 **	--		
10. HCC t0	−0.17	−0.19	−0.14	0.14	0.06	0.09	0.26	0.11	0.23	--	
11. HCC t1	0.09	0.07	0.12	0.15	0.05	0.07	0.20	0.21	0.21	0.52 **	--
*M*	24.07	24.26	24.02	83.55	81.72	85.06	126.38	126.87	129.50	7.83	5.77
*SD*	3.22	3.27	3.36	11.14	11.26	11.36	14.91	13.69	15.87	7.62	4.23

Note. t0: N = 42; t1: N = 39; t2: N = 36; HCC t0: N = 39; HCC t1: N = 36. * *p* < 0.05, ** *p* < 0.01, two-tailed.

**Table 5 ijerph-20-05760-t005:** Intercorrelations of allostatic load and psychological strain at t0, t1, and t2.

Variable	BMI t0	BMI t1	BMI t2	DBP t0	DBP t1	DBP t2	SBP t0	SBP t1	SBP t2	HCC t0	HCC t1
VE t0	−0.24	−0.29	−0.15	−0.19	−0.06	0.08	−0.21	−0.17	0.08	0.10	−0.06
VE t1	−0.28	−0.34 *	−0.22	−0.22	−0.13	−0.04	−0.23	−0.26	−0.02	0.03	−0.19
VE t2	−0.25	−0.29	−0.17	−0.25	−0.13	0.04	−0.34 *	−0.27	−0.11	−0.07	−0.18
CW t0	−0.26	−0.31	−0.15	−0.01	0.15	0.23	−0.07	0.03	0.24	0.09	0.01
CW t1	−0.18	−0.23	−0.10	0.02	0.11	0.22	−0.02	0.05	0.21	0.22	0.03
CW t2	−0.08	−0.14	−0.03	−0.05	−0.01	0.24	−0.24	−0.17	0.04	0.12	0.04
WO t0	−0.10	−0.10	0.09	0.01	0.08	0.29	0.00	0.03	0.36 *	0.10	−0.07
WO t1	−0.25	−0.29	−0.14	−0.18	−0.15	0.03	−0.23	−0.15	0.08	0.08	−0.09
WO t2	−0.12	−0.12	0.04	−0.06	−0.01	0.23	−0.12	−0.07	0.20	0.07	−0.08
OP t0	−0.25	−0.32 *	−0.19	0.07	−0.01	0.08	0.07	−0.04	0.22	0.20	−0.07
OP t1	−0.29	−0.33 *	−0.18	−0.11	−0.06	0.12	−0.12	−0.10	0.11	0.24	0.02
OP t2	−0.22	−0.25	−0.11	−0.11	−0.06	0.22	−0.21	−0.13	0.11	0.13	−0.06
PP t0	−0.19	−0.22	−0.12	0.01	0.15	0.27	−0.13	−0.04	0.22	−0.09	−0.26
PP t1	−0.22	−0.26	−0.16	−0.08	0.18	0.20	−0.17	−0.05	0.14	0.11	−0.27
SP t0	−0.23	−0.24	−0.10	0.00	0.03	0.16	0.07	−0.03	0.19	0.10	−0.10
SP t1	−0.13	−0.17	0.06	0.00	0.05	0.20	−0.05	−0.03	0.14	0.14	−0.15
SP t2	−0.12	−0.15	−0.06	−0.05	0.03	0.27	−0.21	−0.09	0.09	0.07	−0.12
FP t0	0.10	0.12	0.24	0.27	0.16	0.32	0.10	0.14	0.30	0.08	0.19
FP t1	0.21	0.15	0.25	0.37 *	0.22	0.26	0.31	0.26	0.31	−0.04	−0.05
FP t2	0.30	0.30	0.38 *	0.42 **	0.23	0.40 *	0.09	0.21	0.19	−0.08	0.15

Note. * *p* < 0.05, ** *p* < 0.01, two-tailed. Psychological strain: t0: N = 42; t1 and t2: N = 39. Physiological variables: t0: N = 42; t1: N = 39; t2: N = 36; HCC t0: N = 39; HCC t1: N = 36. VE = vital exhaustion; CW = chronic worry; WO = work overload; OP = occupational problems; PP = physical problems; SP = self-related problems; FP = family-related problems.

**Table 6 ijerph-20-05760-t006:** Overview of LASSO-regularized regression coefficients.

Independent Variables		Psychological Strain							Allostatic Load			
		VE	CW	WO	OP	PP	SP	FP	BMI	SBP	DBP	HCC
Individual factors	Core self-evaluation	−0.23	−0.16		−0.09	−0.13	−0.06					
	Teacher self-efficacy											
	Neuroticism	0.08	0.19	0.2	0.18	0.07	0.08					
	Extraversion								0.07			
	Personal competence											
	Life satisfaction			−0.1	−0.3		−0.15					
	Work satisfaction											
	Satisfaction with oneself					−0.06						
	Satisfaction with family											
	Satisfaction with friends											
	Coping styles											
	Excessive work engagement			0.11								
	Striving for perfection											
	Sense of achievement											
	Professional ambition											
	Significance of work								0.07			
	Resignation tendency					0.09			−0.19			
	Offensive problem solving					−0.06						
	Ability to distance oneself					−0.13	−0.1			−0.11		
	Inner peace and balance								0.17			
	Seeking positive experiences			−0.19							−0.13	
Organizational factors	Total lessons								−0.05			
	Observed classroom environment											
	Student aggressions								−0.05			
	Setting disruptions											
	Teacher–student relationship									−0.06		
	Classroom management											
Transactional factors	Perceived classroom environment											
	Student disruptions		0.26	0.14	0.23	0.19	0.15		0.18			
	Student aggressions											
	Setting disruptions	0.11			0.1							
	Classroom management					0.21			−0.13			
	Teacher–student relationship											
	Perceived social support											
	Experience of social support					−0.2						
	Instrumental support											
	from other teachers								−0.18	−0.1	−0.11	
	from school administration								−0.12	−0.17	−0.14	
	Perceived social stressors											
	Fear of negative evaluation											
	Lack of social recognition											
	Social isolation											
	Social tensions								0.07			

Note. VE = vital exhaustion; CW = chronic worrying; WO = work overload; OP = occupational problems; PP = physical problems; SP = self-related problems; FP = family-related problems; BMI = body mass index; SBP = systolic blood pressure; DBP = diastolic blood pressure; HCC = hair cortisol concentration.

## Data Availability

The data presented in this study are available upon request from the corresponding author. The data are not publicly available due to privacy concerns.

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
