# Peer review of "Predictors of Psychological Strain and Allostatic Load in Teachers: Examining the Long-Term Effects of Biopsychosocial Risk and Protective Factors Using a LASSO Regression Approach"

_ijerph, 2023, doi:10.3390/ijerph20105760_

Round 1

Reviewer 1 Report

I enjoyed reviewing this paper. This study examined which factors predicted teachers’ psychological strain and allostatic load over two years. However, there are several major issues in this paper to be addressed, especially in research motivation, theoretical backgrounds, and theoretical implications. Some comments are listed below:

1. The introduction needs to be further improved. Authors should further articulate the relationship between these core constructs based on a theoretical perspective, and a complete framework of the study is not seen in the introduction. The research gap should be thoroughly addressed. What we have known? What we don’t know? What need to do? So what? In addition, the contributions of this study should be stated briefly on the end of introduction. 

2. This study explored three categories of stress factors, including individual, organizational, and transactional. However, authors essentially listed these factors without explaining the relationship between these factors and teacher stress based on relevant theory.

3. The study included 42 teachers (28 female, Mage = 39.66, SD = 11.99) at baseline. After the first measurement, one teacher moved abroad and two teachers withdrew their participation due to pregnancy, resulting in a sample of 39 teachers two years later. Therefore, I am more concerned about whether small samples can draw reliable conclusions.

4. The discussion about the results looks somewhat redundant. This part should be refined and concluded, explaining the most interesting and meaningful results. Conversely, the illustration of theoretical implication is not adequate. The authors should make in-depth analysis of theoretical contributions and its expansion of existing literature.

Reviewer 2 Report

This is a very important study related to stress and physical/physiological health of teachers. It appears well documented and well conducted. There are only very few rather formal aspects which can/must be improved, and I attach the corresponding file for reference.

Author Response

Thank you very much, we have updated the manuscript and implemented the named aspects where possible.

  • Unfortunately, hyphenation is automatically enabled in the provided template, and we cannot remove it from the title
  • Regarding the spelling of Martin grosse Holtforth, grosse is correctly spelled in all lowercase letters
  • The wording in the abstract has been updated as proposed
  • Table 1 has been replaced by Figure 1, where bullet points are closer to the terms
  • Subchapter titles have been suppressed

Round 2

Reviewer 1 Report

The authors have made in-depth revisions to the article and the quality of the article has been improved.